# Prescription of benzodiazepines, z-drugs, and gabapentinoids and mortality risk in people receiving opioid agonist treatment: Observational study based on the UK Clinical Practice Research Datalink and Office for National Statistics death records

**John Macleod**[1]*, **Colin Steer**[1], **Kate Tilling**[1], **Rosie Cornish**[1], **John Marsden**[2], **Tim Millar**[3], **John Strang**[2], **Matthew Hickman**[1]

**1** Population Health Sciences, Bristol Medical School, Faculty of Health Sciences, University of Bristol, Bristol, United Kingdom, **2** Addictions Department, Institute of Psychiatry, Psychology & Neuroscience, King's College London, London, United Kingdom, **3** Centre for Mental Health & Safety, School of Health Sciences, University of Manchester, Manchester, United Kingdom

\* john.macleod@bristol.ac.uk

## Abstract

### Background

Patients with opioid dependency prescribed opioid agonist treatment (OAT) may also be prescribed sedative drugs. This may increase mortality risk but may also increase treatment duration, with overall benefit. We hypothesised that prescription of benzodiazepines in patients receiving OAT would increase risk of mortality overall, irrespective of any increased treatment duration.

### Methods and findings

Data on 12,118 patients aged 15–64 years prescribed OAT between 1998 and 2014 were extracted from the Clinical Practice Research Datalink. Data from the Office for National Statistics on whether patients had died and, if so, their cause of death were available for 7,016 of these patients. We identified episodes of prescription of benzodiazepines, z-drugs, and gabapentinoids and used linear regression and Cox proportional hazards models to assess the associations of co-prescription (prescribed during OAT and up to 12 months post-treatment) and concurrent prescription (prescribed during OAT) with treatment duration and mortality. We examined all-cause mortality (ACM), drug-related poisoning (DRP) mortality, and mortality not attributable to DRP (non-DRP). Models included potential confounding factors. In 36,126 person-years of follow-up there were 657 deaths and 29,540 OAT episodes, of which 42% involved benzodiazepine co-prescription and 29% concurrent prescription (for z-drugs these respective proportions were 20% and 11%, and for gabapentinoids 8% and 5%). Concurrent prescription of benzodiazepines was associated with

**Data Availability Statement:** The data underlying the results presented in the study are available from the Clinical Practice Research Datalink: https://www.cprd.com/.

**Funding:** The study was supported by NIHR HS&DR (https://www.nihr.ac.uk/funding-and-support/funding-for-research-studies/funding-programmes/health-services-and-delivery-research/) Project: 12/136/105 - Evaluating the impact of opiate substitution treatment on drug related deaths in the population: a natural experiment using primary care, other drug treatment databases & model projections. ISAC CPRD Protocol 14_073R2, principal investigator MH.

**Competing interests:** I have read the journal's policy and the authors of this manuscript have the following competing interests: In the past three years, JMar declares research grants from the National Institute for Health Research (NIHR; randomised controlled trial of depot naltrexone for OUD, and a randomised controlled trial of acamprosate for alcohol use disorder); and the NIHR Biomedical Research Centre for Mental Health at South London and Maudsley NHS Mental Health Foundation Trust (SLaM; randomised controlled trial of novel cognitive therapy for cocaine use disorder). He has worked part-time as the Senior Academic Advisor for the Alcohol, Drugs, Tobacco and Justice Division, Health and Wellbeing Directorate, PHE and he is a clinical academic consultant for the US National Institute on Drug Abuse, Centre for Clinical Trials Network. JM declares an unrestricted research grant at IoPPN and SLaM from Indivior via Action on Addiction for a completed randomised controlled trial of personalised psychosocial intervention in opioid agonist medication for opioid use disorder and, with MK, unrestricted research grant funding at IoPPN and SLaM from Indivior for a three-year, multi-centre, randomised controlled trial of injectable depot buprenorphine for opioid use disorder (2019-2021). He has received honoraria and travel support from Reckitt-Benckiser (2016; treatment of OUD and PCM Scientific and Martindale for the Improving Outcomes in Treatment of Opioid Dependence conference (2018; contribution and chairing). TM has received research funding from the UK National Treatment Agency for Substance Misuse, Public Health England, the Home Office, and Change Grow Live, a 3rd-sector provider of substance misuse services. He has been a member of the organising committee for conferences supported by unrestricted educational grants from Reckitt Benckiser, Lundbeck, Martindale Pharma, and

increased duration of methadone treatment (adjusted mean duration of treatment episode 466 days [95% CI 450 to 483] compared to 286 days [95% CI 275 to 297]). Benzodiazepine co-prescription was associated with increased risk of DRP (adjusted HR 2.96 [95% CI 1.97 to 4.43], $p < 0.001$), with evidence of a dose–response effect, but showed little evidence of an association with non-DRP (adjusted HR 0.91 [95% CI 0.66 to 1.25], $p = 0.549$). Co-prescription of z-drugs showed evidence of an association with increased risk of DRP (adjusted HR 2.75 [95% CI 1.57 to 4.83], $p < 0.001$) but little evidence of an association with non-DRP (adjusted HR 0.79 [95% CI 0.49 to 1.28], $p = 0.342$). There was no evidence of an association of gabapentinoid co-prescription with DRP (HR 1.54 [95% CI 0.60 to 3.98], $p = 0.373$) but evidence of an association with increased non-DRP (HR 1.83 [95% CI 1.28 to 2.62], $p = 0.001$). Concurrent benzodiazepine prescription also increased mortality risk after consideration of duration of OAT (adjusted HR for DRP with benzodiazepine concurrent prescription 3.34 [95% CI 2.14 to 5.20], $p < 0.001$). The main limitation of this study is the possibility that unmeasured confounding factors led to an association between benzodiazepine prescription and DRP that is not causal.

## Conclusions

In this study, co-prescription of benzodiazepine was specifically associated with increased risk of DRP in opioid-dependent individuals. Co-prescription of z-drugs and gabapentinoids was also associated with increased mortality risk; however, for z-drugs there was no evidence for a dose–response effect on DRP, and for gabapentinoids the increased mortality risk was not specific to DRP. Concurrent prescription of benzodiazepine was associated with longer treatment but still increased risk of death overall. Clinicians should be cautious about prescribing benzodiazepines to opioid-dependent individuals.

## Author summary

### Why was this study done?

- Deaths in opioid-dependent individuals are increasing despite the fact that many receive opioid agonist therapy, an effective treatment.

- It is possible that medicines prescribed in addition to opioid agonist treatment may influence risk of death, even if these medicines also increase retention in treatment.

### What did the researchers do and find?

- Using a large database of medical records from UK primary care linked to death records, the researchers examined both risk of death and treatment duration amongst individuals receiving opioid agonist treatment according to whether they were also prescribed certain sedative medicines.

Britannia Pharmaceuticals Ltd, for which he received no personal remuneration. He is a member of the UK Advisory Council on the Misuse of Drugs. JS is a clinician and researcher and has worked extensively with agencies in the addiction treatment fields and addiction-related charities and with government departments and has contributed to clinical guidelines on treatment types and provision. JS's employer (King's College London) has received, connected to his work, project grant support and/or honoraria and/or consultancy payments from Department of Health, NTA (National Treatment Agency), PHE (Public Health England), Home Office, NICE (National Institute for Health and Clinical Excellence), and EMCDDA (European Monitoring Centre for Drugs and Drug Addiction) as well as research grants from (last 3 years) NIHR (National Institute on Health Research), MRC (Medical Research Council) and Pilgrim Trust. He has also worked with WHO (World Health Organization), UNODC (United Nations Office on Drugs and Crime), EMCDDA, FDA (US Food and Drug Administration) and NIDA (US National Institute on Drug Abuse) and also other international government agencies. JS's employer (King's College London) has also received, connected to his work, research grant support and/or payment of honoraria, consultancy payments and expenses from pharmaceutical companies (including, past 3 years, Martindale, Indivior, MundiPharma, Braeburn/Camurus) and trial medication supply from iGen and Braeburn. JS's employer (King's College London) has registered intellectual property on an innovative buccal naloxone with which JS is involved, and JS has been named in a patent registration by a Pharma company as inventor of a potential concentrated naloxone nasal spray. For updated information see http://www.kcl.ac.uk/ioppn/depts/addictions/people/hod.aspx. MH acknowledges support from NIHR Health Protection Research Unit in Evaluation of Interventions and the NIHR School of Public Health Research. MH has received unrelated unrestricted honoraria from Gilead, Abbvie, Jansen and Merck Serono. No other disclosures by the other authors are reported.

**Abbreviations:** ACM, all-cause mortality; CPRD, Clinical Practice Research Datalink; DRP, drug-related poisoning; OAT, opioid agonist treatment; ONS, Office for National Statistics.

- They found that individuals additionally prescribed a particular type of sedative—benzodiazepines—had increased risk of death from overdose, despite staying in treatment longer.

- This pattern wasn't seen with the other additional prescribed medicines considered.

### What do these findings mean?

- These findings may mean that, in individuals dependent on opioids, taking benzodiazepines may cause an increased risk of death from overdose.

- If this is true, then prescribing benzodiazepines to opioid-dependent individuals should be avoided.

- It may be that amongst individuals receiving opioid agonist treatment, those prescribed benzodiazepines are different from those who don't receive a benzodiazepine prescription in ways we were unable to consider, and that it is some aspect of this difference that increases their risk of death from overdose.

## Introduction

Deaths amongst users of illicit opioids are increasing despite the fact that in many jurisdictions, such as the UK, a high proportion of users are in treatment [1]. Evidence suggests that opioid substitution therapy (OAT) is effective in improving health and, in individuals on long-term maintenance therapy, in reducing mortality risk [2–6]. Though mortality rates are highest amongst users not in treatment, deaths during treatment and following the end of treatment are still observed [6,7]. Several aspects of treatment may influence this mortality risk, and clinical guidelines emphasise these [8]. Guidelines typically advise against additional prescription of drugs that may potentiate the respiratory depressive effects of opioids and increase risk of overdose death [9]. These drugs include benzodiazepines, "z-drugs" (zaleplon, zolpidem, and zopiclone), and gabapentinoids (gabapentin and pregabalin).

Despite this advice, these drugs are commonly prescribed to opioid-dependent individuals [10–12]. This practice may reflect both uncertainty around the strength of existing evidence that the practice is harmful and a belief that additional prescription may have benefits through increasing treatment retention that have not been investigated yet [13].

We studied the association between additional prescription and treatment retention and the question of whether additional prescription was associated with overall benefit or harm in opioid-dependent individuals receiving OAT using data from a UK database of electronic patient records from primary care linked to a national mortality registry. We examined associations of concurrent prescription (additional prescribing during OAT) and co-prescription (additional prescribing during OAT and up to 1 year after OAT) of benzodiazepines, z-drugs, and gabapentinoids with treatment duration and mortality. We hypothesised that both concurrent and co-prescription of these drugs would be associated with increased mortality risk, despite any effect on treatment duration. We further hypothesised that a causal basis for such increased risk would be reflected in a specific association of both concurrent and co-prescription with drug-related poisonings (DRPs), a dose–response association of greater risk with

higher doses of the additional drug, and an effect estimate robust to adjustment for possible confounding.

## Methods

### Study protocol and pre-specified analyses

Our study protocol was reviewed by the Independent Scientific Advisory Committee of the Clinical Practice Research Datalink (CPRD) (S1 Protocol). Our full study report including pre-specified analyses is available at https://www.journalslibrary.nihr.ac.uk/hsdr/hsdr07030#/abstract.

### Ethical approval

Research studies approved by the CPRD Independent Scientific Advisory Committee do not require additional ethical approval. Scientific approval for this study was given by the CPRD Independent Scientific Advisory Committee as above, and no additional informed consent was required as there was no individual patient involvement The permissions covering the research use of CPRD data are described at https://www.cprd.com/transparency-information.

### OAT patients

Data were identified for 49,279 primary care patients within the CPRD (https://www.cprd.com/home/) who had received methadone or buprenorphine between the study dates of 1 January 1998 and 31 July 2014. CPRD is a large database of anonymised patient records from 674 general practices and over 11 million patients in the UK, and currently collects data from approximately 7% of the UK population [14]. As described elsewhere, we used diagnostic and prescription formulation information to exclude 26,324 patients who were prescribed buprenorphine or methadone for pain relief, as well as 9,950 patients who received doses below the minimum expected for OAT (i.e., at least 20 mg methadone or 4 mg buprenorphine per day) [15]. We also excluded patients who were aged <15 or >64 years at the start of the study based on an assumption that their opioid prescription was less likely to be for OAT. This left a total of 667,288 prescriptions related to 12,118 patients; 65% were only prescribed methadone, 22% only buprenorphine, and 13% both medications (S1 Patient Flowchart).

### OAT episodes

For each individual, we classified their follow-up time into episodes on and off OAT. These covered the period on treatment and the period off treatment up to 1 year after the prescription ended. An individual could experience several consecutive episodes of OAT. We defined a new episode of OAT where a gap of >28 days existed between the expected completion of one prescription and the start of the next, or, in other words, where the prescription interval exceeded the prescription duration by more than 28 days.

First (observed) episodes were potentially subject to left censoring because of the study start date, date of registration with the current practice, or the CPRD up-to-standard date. All last episodes were right censored because of the study end date, the last collection date by CPRD, the patient leaving the CPRD practice, death of the patient, or censoring 1 year after cessation of the last treatment. The earliest of these 5 events defined the end of follow-up. We censored follow-up at 1 year after the end of the last treatment episode to avoid the dilution of mortality risks where patients may have been at lower risk due to the possibility of recovery.

## Additionally prescribed medications

Ten benzodiazepine and 3 z-drug medications were listed in British National Formulary, with chapters 4.1 and 4.2 providing information on recommended doses [16]. There were 365,582 prescriptions for the 10 benzodiazepines, 75,926 for the 3 z-drugs, and 23,451 for the 2 gabapentinoids. Treatment episodes were generated in a similar fashion as for OAT except that the minimum gap was reduced to 14 days for benzodiazepine and z-drugs, reflecting their shorter recommended treatment duration.

## Confounding factors

Adjustment was made for sex, age, year of treatment, comorbidity, UK region, type of OAT, and OAT period. Comorbidity was calculated based upon 17 chronic illnesses as defined by Khan et al. [17]. The 3,156 READ codes were translated to the current CPRD medcodes. This was possible for 2,856 codes. Occurrence of the medcodes within the clinical notes generated event dates for time-varying covariates. Each occurrence received a weighting from 1 to 6 related to the particular chronic illness. The weights were summed across time and illnesses to create the index. To take account of associations between benzodiazepine, z-drugs, and gabapentinoids, all other drug exposures were included in models examining the effects of each main drug.

## Deaths

All-cause mortality (ACM) was derived from dates recorded in the clinical notes and from patient records within CPRD. Dates of death were also available from Office for National Statistics (ONS) records for a subset of the CPRD patients in England. ONS data included details of primary and secondary causes of death, allowing DRP to be defined (S1 Table). The ICD-9/ICD-10 codes assigned in our study are based upon ONS but extended to take account of possible underreporting in the use of codes relating to non-specific or unknown causes [18]. In addition, since deaths from DRP contribute to ACM, we examined deaths not attributable to DRP as a separate category (non-DRP).

## Statistical analyses

We investigated the association of concurrent prescription with OAT duration using linear regression. As duration of treatment episodes was highly positively skewed, we repeated this analysis after undertaking a log transformation of duration.

We then used Cox proportional hazards survival analysis to analyse mortality risk associated with co-prescription of benzodiazepines, z-drugs, or gabapentinoids, both on and off OAT (i.e., within a treatment episode) in unadjusted and adjusted analyses. Where a main effect of co-prescription was suggested by these analyses, we investigated whether this effect was modified by an interaction with OAT period (i.e., on or off treatment). For benzodiazepine or z-drug dose effects (2 degrees of freedom), a linear trend was also fitted to examine the effect of dose, with "high" dose defined as a daily dose above the maximum recommended in the British National Formulary (S2 Table). Because of smaller numbers, gabapentinoids were assessed as off/on treatment only.

We then assessed the overall effect of concurrent prescription on mortality risk, using Cox proportional hazards survival analysis. This analysis effectively allows for the longer OAT duration associated with concurrent prescription. As sensitivity analyses, we repeated the main analyses excluding the first treatment episode (to address left censoring).

All analyses were undertaken using Stata version 14.2. We assessed the assumption of constant hazards over time by dividing each episode into weeks 0–4 on treatment, the rest of the period on treatment, weeks 0–4 off treatment, and the rest of the period off treatment. As a further sensitivity analysis, we repeated the analyses using Poisson regression. We also repeated our survival analysis of the effects of co-prescription on risk of DRP considering competing risk of death from other causes.

For clarity we include a summary of our analyses and the relevant Stata commands as S1 Appendix.

## Results

### Data characteristics

These data contained 29,540 OAT episodes: 17,391 with methadone (median treatment duration, 0.30 years; range, 0.00–23.00; interquartile range, 0.06–1.07 years), 9,180 with buprenorphine (median treatment duration, 0.11 years; range, 0.00–17.00; interquartile range, 0.02–0.38), and 2,969 episodes relating to more than 1 drug. Total follow-up time was 36,126 person-years, during which there were 17,139 episodes of benzodiazepine treatment, 5,957 episodes of z-drug treatment, and 2,822 episodes of gabapentinoid treatment. Benzodiazepines were prescribed during 10,022 OAT episodes in 7,059 patients. Z-drugs were prescribed during 4,279 OAT episodes in 2,822 patients. Gabapentinoids were prescribed during 1,827 OAT episodes in 1,281 patients.

### Study participants

Characteristics of study participants are summarised in Table 1. Two-thirds of the sample were male, and mean age of the sample at study exit was 39 years, with a mean follow-up per patient of 3.4 years. During the observation period, 42% of individuals in the sample were co-prescribed benzodiazepines, 20% were co-prescribed z-drugs, and 8% were co-prescribed gabapentinoids.

**Table 1. Description of study participants overall and by cause of death.**

| Characteristic | CPRD | | CPRD linked to ONS | | |
|---|---|---|---|---|---|
| | Total | All-cause deaths | Total | Drug-related poisoning deaths | Non-drug/other causes of death |
| Patients (n) | 12,118 | 657 | 7,106 | 113 | 285 |
| Male (%) | 67.3 | 65.1 | 68.1 | 87.6 | 61.4 |
| Mean (SD) age at exit, years | 38.8 (10.4) | 47.3 (12.3) | 39.3 (10.7) | 37.0 (9.6) | 51.9 (11.6) |
| Median (IQR) follow-up, years | 2.07 (1.03–4.79) | 0.89 (0.29–3.25) | 2.09 (1.03–4.82) | 1.90 (0.66–4.62) | 0.78 (0.20–2.49) |
| Comorbid at exit (%) | 31.4 | 66.2 | 32.8 | 39.8 | 76.1 |
| **Ever prescribed (%)** | | | | | |
| Benzodiazepine | 46.8 | 58.3 | 45.0 | 65.5 | 52.3 |
| Z-drug | 24.0 | 23.6 | 26.4 | 38.9 | 23.2 |
| Gabapentinoid | 8.8 | 26.5 | 9.0 | 8.8 | 33.0 |
| **Concurrent with OAT (%)** | | | | | |
| Benzodiazepine | 42.2 | 52.5 | 40.2 | 61.9 | 47.0 |
| Z-drug | 19.7 | 20.7 | 21.8 | 31.0 | 22.5 |
| Gabapentinoid | 7.6 | 25.4 | 7.7 | 8.8 | 31.6 |

CPRD, Clinical Practice Research Datalink; OAT, opioid agonist treatment; ONS, Office for National Statistics.

Unadjusted mortality rates according to participant characteristics other than co-prescription are summarised in S3 Table. DRP was higher in men, not significantly associated with age, higher in the observation period up to 2004, and higher in patients with more comorbidity. Non-drug-related (non-DRP) mortality was higher in women, increased with increasing participant age and comorbidity, and did not show any evidence of trend across the observation period. All mortality rates showed considerable regional variation.

### Prevalence of co-prescription over time

We examined prevalence of co-prescription amongst patients on OAT between 1998 and 2014. Benzodiazepine co-prescription decreased over this period from 43% (95% CI 40% to 46%) in 1998 to 28% (95% CI 26% to 29%) in 2014. In contrast, co-prescription of z-drugs remained constant at 13% (95% CI 11% to 15%), while co-prescription of gabapentinoids increased from 0.5% (95% CI 0.1% to 1.1%) in 1998 to 12% (95% CI 11% to 14%) in 2014.

### Association of concurrent prescription of benzodiazepines, z-drugs, and gabapentinoids with duration of OAT episodes

In total, 27,598 of 29,540 episodes of OAT were available for this analysis. During 2,969 of these episodes, more than 1 substitute drug was prescribed: We included 1,027 episodes where methadone and buprenorphine were prescribed consecutively but excluded 1,942 episodes where other opioid substitutes such as dihydrocodeine were prescribed. Of these 27,598 OAT episodes, 17,111 (62%) had no concurrent prescription. Buprenorphine episodes were considerably shorter than methadone episodes (Table 2). Concurrent prescription of both benzodiazepines and z-drugs was associated with a similar proportional increase in duration of both methadone and buprenorphine treatment episodes (adjusted mean duration of OAT episode without concurrent prescription of benzodiazepine 244 [95% CI 236 to 252] days, with concurrent prescription of benzodiazepine 416 [95% CI 404 to 429] days, with concurrent

**Table 2. Comparison of median, unadjusted mean, and adjusted mean opioid agonist treatment (OAT) duration with concurrent prescription of benzodiazepines, z-drugs, and gabapentinoids, by OAT type.**

| OAT prescribed | Concurrent prescription | Number of episodes[a] | Median duration | Unadjusted mean duration (95% CI) | Adjusted[b] mean duration (95% CI) |
|---|---|---|---|---|---|
| Any OAT[c] | None | 17,111 | 62 | 231 (223 to 240) | 244 (236 to 252) |
| | Benzodiazepine | 7,961 | 147 | 423 (410 to 437) | 416 (404 to 429) |
| | Z-drug | 3,165 | 223 | 448 (427 to 469) | 441 (421 to 461) |
| | Gabapentinoid | 1,405 | 86 | 270 (239 to 301) | 189 (159 to 219) |
| Methadone | None | 10,662 | 84 | 272 (261 to 284) | 286 (275 to 297) |
| | Benzodiazepine | 5,283 | 182 | 471 (453 to 489) | 466 (450 to 483) |
| | Z-drug | 1,930 | 277 | 491 (462 to 520) | 483 (456 to 511) |
| | Gabapentinoid | 745 | 125 | 346 (301 to 392) | 224 (180 to 268) |
| Buprenorphine | None | 5,983 | 32 | 131 (121 to 140) | 135 (126 to 144) |
| | Benzodiazepine | 2,250 | 59 | 231 (214 to 248) | 234 (217 to 250) |
| | Z-drug | 953 | 93 | 270 (244 to 296) | 266 (240 to 291) |
| | Gabapentinoid | 628 | 52 | 180 (149 to 212) | 140 (109 to 171) |

Duration is reported as days.

[a]Includes OAT episodes where more than 1 drug was concurrently prescribed. For both types, this involved 2,044 episodes.

[b]Adjusted for sex, age, year, comorbidity, region, and, where applicable, OAT type and concurrent prescription of benzodiazepine, z-drug, or gabapentinoid.

[c]Includes episodes of only methadone, only buprenorphine, or both medications consecutively prescribed.

**Table 3. Comparison of log opioid agonist treatment (OAT) duration, plus interquartile range (IQR) of duration, for concurrent prescription of benzodiazepines, z-drugs, and gabapentinoids, by OAT type.**

| OAT prescribed | Concurrent prescription | Unadjusted | Adjusted[a] | IQR |
|---|---|---|---|---|
| Any OAT[b] | None | Ref | Ref | 13–230 |
| | Benzodiazepine | 0.64 (0.56, 0.71) | 0.48 (0.42, 0.54) | 27–568 |
| | Z-drug | 0.88 (0.79, 0.98) | 0.77 (0.69, 0.85) | 53–688 |
| | Gabapentinoid | 0.13 (−0.01, 0.27) | 0.23 (0.10, 0.36) | 18.5–362.5 |
| Methadone | None | Ref | Ref | 18–308 |
| | Benzodiazepine | 0.63 (0.56, 0.71) | 0.49 (0.42, 0.55) | 44–712.5 |
| | Z-drug | 0.91 (0.82, 1.00) | 0.78 (0.69, 0.86) | 84–878 |
| | Gabapentinoid | 0.21 (−0.002, 0.43) | 0.20 (0.02, 0.38) | 29–629 |
| Buprenorphine | None | Ref | Ref | 9–139 |
| | Benzodiazepine | 0.63 (0.50, 0.76) | 0.43 (0.32, 0.54) | 14–382.5 |
| | Z-drug | 0.97 (0.82, 1.13) | 0.64 (0.50, 0.78) | 35–524.5 |
| | Gabapentinoid | 0.16 (−0.02, 0.34) | 0.30 (0.13, 0.47) | 13–226 |

Treatment episodes as in Table 2.

[a]Adjusted for sex, age, year, comorbidity, region, and, where applicable, OAT type and concurrent prescription of benzodiazepine, z-drug, or gabapentinoid.

[b]Includes episodes of only methadone, only buprenorphine, or both medications consecutively prescribed.

prescription of z-drug 441 [95% CI 421 to 461] days, and with concurrent prescription of gabapentinoid 189 [95% CI 159 to 219] days).

Duration of OAT episodes was highly positively skewed; because of this we undertook a log transformation prior to our analysis and also report interquartile ranges (IQRs) of treatment duration in addition to the median (Table 3).

## Associations of co-prescription with mortality

**Benzodiazepines.** There was strong evidence of a higher rate of DRP during periods of benzodiazepine co-prescription (adjusted HR 2.96 [95% CI 1.97 to 4.43], $p < 0.001$), with evidence of a dose–response relationship (adjusted HR on normal dose 2.51 [95% CI 1.57 to 4.01], $p < 0.001$; on high dose 4.57 [95% CI 2.46 to 8.47], $p < 0.001$). There was little evidence of a difference in rates of ACM and non-DRP according to whether patients were on or off benzodiazepines (Table 4).

**Z-drugs and gabapentinoids.** No significant association between co-prescribed z-drugs and ACM was apparent in adjusted analyses (Table 4). Co-prescribed z-drugs showed an overall significant association with DRP but no dose–response effect (adjusted HR for DRP 3.66 [95% CI 1.86 to 7.19], $p = 0.001$, for normal dose; 1.55 [95% CI 0.59 to 4.06], $p = 0.001$, for high dose). Gabapentinoid co-prescription was significantly associated with non-DRP but not with DRP (DRP adjusted HR 1.54 [95% CI 0.60 to 3.98], $p = 0.37$; non-DRP adjusted HR 1.83 [95% CI 1.28 to 2.62], $p = 0.001$). It was not possible to examine evidence for a dose–response association of mortality with gabapentinoids.

## Interactions between benzodiazepine co-prescription and OAT period

There was a strong and significant association between co-prescribed benzodiazepine and increased risk of death that appeared specific to DRP. We found no evidence of any interaction of this effect with treatment period (Table 5).

**Table 4. Mortality rates and hazard ratios for all-cause mortality, DRP, and other causes of death by co-prescription exposure to benzodiazepines, z-drugs, or gabapentinoids in people with opioid dependency in primary care.**

| Co-prescription | Deaths | PY | MR (per 100 PY) | Unadjusted | | Adjusted[a] | |
|---|---|---|---|---|---|---|---|
| | | | | HR (95% CI) | p-Value | HR (95% CI) | p-Value |
| *All-cause mortality* | | | | | | | |
| **Benzodiazepine** | | | | | | | |
| Off | 513 | 28,766 | 1.78 | 1 (ref) | 0.718 | 1 (ref) | 0.105 |
| On | 144 | 7,361 | 1.96 | 1.03 (0.86 to 1.25) | | 1.17 (0.97 to 1.42) | |
| Off | 513 | 28,766 | 1.78 | 1 (ref) | 0.595 | 1 (ref) | 0.263 |
| On normal dose | 116 | 5,909 | 1.96 | 1.00 (0.82 to 1.22) | | 1.18 (0.96 to 1.46) | |
| On high dose | 28 | 1,452 | 1.93 | 1.22 (0.83 to 1.78) | | 1.12 (0.75 to 1.66) | |
| *Linear effect of dose* | — | — | — | 1.05 (0.91 to 1.22) | 0.507 | 1.11 (0.96 to 1.29) | 0.158 |
| **Z-drug** | | | | | | | |
| Off | 606 | 34,264 | 1.77 | 1 (ref) | 0.014 | 1 (ref) | 0.124 |
| On | 51 | 1,862 | 2.74 | 1.43 (1.08 to 1.91) | | 1.26 (0.94 to 1.69) | |
| Off | 606 | 34,264 | 1.77 | 1 (ref) | 0.048 | 1 (ref) | 0.280 |
| On normal dose | 29 | 1,008 | 2.88 | 1.42 (0.98 to 2.06) | | 1.21 (0.83 to 1.78) | |
| On high dose | 22 | 854 | 2.58 | 1.45 (0.95 to 2.22) | | 1.34 (0.86 to 2.11) | |
| **Gabapentinoid** | | | | | | | |
| Off | 574 | 35,129 | 1.63 | 1 (ref) | <0.001 | 1 (ref) | <0.001 |
| On | 83 | 998 | 8.32 | 2.79 (2.20 to 3.54) | | 1.71 (1.33 to 2.20) | |
| *DRP* | | | | | | | |
| **Benzodiazepine** | | | | | | | |
| Off | 74 | 16,270 | 0.45 | 1 (ref) | <0.001 | 1 (ref) | <0.001 |
| On | 39 | 3,679 | 1.06 | 2.35 (1.59 to 3.47) | | 2.96 (1.97 to 4.43) | |
| Off | 74 | 16,270 | 0.45 | 1 (ref) | <0.001 | 1 (ref) | <0.001 |
| On normal dose | 25 | 2,889 | 0.87 | 1.93 (1.22 to 3.04) | | 2.51 (1.57 to 4.01) | |
| On high dose | 14 | 790 | 1.77 | 3.83 (2.16 to 6.80) | | 4.57 (2.46 to 8.47) | |
| *Linear effect of dose* | — | — | — | 1.95 (1.50 to 2.54) | <0.001 | 2.22 (1.69 to 2.92) | <0.001 |
| **Z-drug** | | | | | | | |
| Off | 98 | 18,838 | 0.52 | 1 (ref) | 0.001 | 1 (ref) | <0.001 |
| On | 15 | 1,110 | 1.35 | 2.52 (1.46 to 4.34) | | 2.75 (1.57 to 4.83) | |
| Off | 98 | 18,838 | 0.52 | 1 (ref) | 0.001 | 1 (ref) | 0.001 |
| On normal dose | 10 | 593 | 1.69 | 3.17 (1.65 to 6.09) | | 3.66 (1.86 to 7.19) | |
| On high dose | 5 | 517 | 0.97 | 1.78 (0.72 to 4.37) | | 1.55 (0.59 to 4.06) | |
| **Gabapentinoid** | | | | | | | |
| Off | 108 | 19,410 | 0.56 | 1 (ref) | 0.212 | 1 (ref) | 0.373 |
| On | 5 | 538 | 0.93 | 1.79 (0.72 to 4.46) | | 1.54 (0.60 to 3.98) | |
| *Non-DRP* | | | | | | | |
| **Benzodiazepine** | | | | | | | |
| Off | 235 | 16,628 | 1.41 | 1 (ref) | 0.302 | 1 (ref) | 0.549 |
| On | 50 | 3,757 | 1.33 | 0.85 (0.63 to 1.16) | | 0.91 (0.66 to 1.25) | |
| Off | 235 | 16,628 | 1.41 | 1 (ref) | 0.497 | 1 (ref) | 0.711 |
| On normal dose | 44 | 3,007 | 1.46 | 0.88 (0.64 to 1.22) | | 0.94 (0.67 to 1.32) | |
| On high dose | 6 | 751 | 0.80 | 0.67 (0.30 to 1.52) | | 0.72 (0.31 to 1.66) | |
| Linear effect of dose | | | | | | | |
| **Z-drug** | 265 | 19,252 | 1.38 | 1 (ref) | 0.266 | 1 (ref) | 0.342 |
| Off | 20 | 1,134 | 1.76 | 1.29 (0.82 to 2.04) | | 0.79 (0.49 to 1.28) | |
| On | 265 | 19,252 | 1.38 | 1 (ref) | 0.470 | 1 (ref) | 0.669 |

*(Continued)*

**Table 4.** (Continued)

| Co-prescription | Deaths | PY | MR (per 100 PY) | Unadjusted | | Adjusted[a] | |
|---|---|---|---|---|---|---|---|
| | | | | HR (95% CI) | p-Value | HR (95% CI) | p-Value |
| Off | 12 | 601 | 2.00 | 1.42 (0.79 to 2.54) | | 0.84 (0.45 to 1.54) | |
| On normal dose | 8 | 532 | 1.50 | 1.14 (0.57 to 2.32) | | 0.76 (0.35 to 1.62) | |
| On high dose | | | | | | | |
| **Gabapentinoid** | 239 | 19,815 | 1.21 | 1 (ref) | <0.001 | 1 (ref) | 0.001 |
| Off | 46 | 571 | 8.06 | 3.37 (2.44 to 4.65) | | 1.83 (1.28 to 2.62) | |

Unadjusted p-values test for differences in mortality rates by co-prescription treatment. High and normal doses are defined in S2 Table.

[a]Adjusted for sex, year, comorbidity, region, OAT type, OAT period, and, where applicable, benzodiazepine, z-drug, and gabapentinoid exposure. Linear trend was applied to ln(IRR). IRR for on high dose was estimated as on normal dose squared. The deviation from linearity for adjusted models for benzodiazepine: $p = 0.4168$ and 0.5352 for all-cause mortality and DRP, respectively.

DRP, drug-related poisoning; HR, hazard ratio; IRR, incidence rate ratio; MR, mortality rate; OAT, opioid agonist treatment; PY, person-years of follow-up.

## Overall association of concurrent prescription with mortality risk

There was no evidence of any beneficial effect of concurrent prescription such as might arise through increased treatment duration. All concurrent prescription was associated with increased risk of all mortality classes (e.g., HR for DRP with concurrent prescription of benzodiazepine 3.34 [95% CI 2.14 to 5.20], $p < 0.001$; concurrent prescription of z-drug 1.64 [95% CI 1.02 to 2.64]; $p < 0.001$) (Table 6).

## Sensitivity analyses

Similar results for all main analyses were seen using Poisson regression instead of Cox proportional hazards regression (S4 Table) and when the first treatment episode was excluded (S5 Table). Consideration of competing risk of death from other causes did not materially change our survival analysis of the effects of co-prescription on risk of DRP (S6 Table).

## Discussion

## Principal findings

We found that the proportion of opioid-dependent patients prescribed a benzodiazepine fell between 1998 and 2014. Nevertheless, in 2014 almost a third of patients on OAT were co-prescribed a benzodiazepine (mainly diazepam). We found strong evidence of substantially increased mortality risk amongst opioid-dependent patients in primary care who were

**Table 5. Survival analysis results for interactions between co-prescribed benzodiazepines and OAT period (on or off treatment) in relation to the effect on risk of drug-related poisoning.**

| OAT period | Co-prescribed benzodiazepine | Deaths | PY | MR (per 100 PY) | Unadjusted | | Adjusted[a] | |
|---|---|---|---|---|---|---|---|---|
| | | | | | HR (95% CI) | p-Value | HR (95% CI) | p-Value |
| On | Off | 24 | 10,091 | 0.24 | 1 (ref) | **0.896** | 1 (ref) | **0.997** |
| | On | 20 | 2,914 | 0.69 | 2.87 (1.58 to 5.20) | 0.001 | 2.92 (1.60 to 5.33) | 0.001 |
| Off | Off | 50 | 6,179 | 0.81 | 1 (ref) | | 1 (ref) | |
| | On | 19 | 764 | 2.49 | 3.02 (1.78 to 5.15) | <0.001 | 2.92 (1.70 to 5.02) | <0.001 |

Interaction p-value shown in bold.

[a]Adjusted for sex, year, comorbidity, region, OAT type, OAT period, and, where applicable, benzodiazepine, z-drug, and gabapentinoid exposure.

HR, hazard ratio; MR, mortality rate; OAT, opioid agonist treatment; PY, person-years of follow-up.

**Table 6. Survival analysis showing overall effect of concurrent exposure to benzodiazepines or z-drugs.**

| Mortality | Concurrent exposure with OAT | Unadjusted | | Adjusted[a] | |
|---|---|---|---|---|---|
| | | HR (95% CI) | *p*-Value | HR (95% CI) | *p*-Value |
| All cause | None | 1 (ref) | 0.052 | 1 (ref) | <0.001 |
| | Benzodiazepine | 1.22 (1.04 to 1.42) | | 1.87 (1.55 to 2.25) | |
| | Z-drug | 0.97 (0.79 to 1.19) | | 1.37 (1.09 to 1.72) | |
| DRP | None | 1 (ref) | 0.001 | 1 (ref) | <0.001 |
| | Benzodiazepine | 1.98 (1.35 to 2.90) | | 3.34 (2.14 to 5.20) | |
| | Z-drug | 1.24 (0.81 to 1.89) | | 1.64 (1.02 to 2.64) | |
| Non-DRP | None | 1 (ref) | 0.570 | 1 (ref) | <0.001 |
| | Benzodiazepine | 1.10 (0.86 to 1.40) | | 1.73 (1.28 to 2.33) | |
| | Z-drug | 1.11 (0.82 to 1.50) | | 1.37 (0.95 to 1.96) | |

Results for gabapentinoids are not reported because there was no evidence of an effect of increased treatment duration.

[a]Adjusted for sex; year; comorbidity; region; OAT type; OAT period; off treatment prescription of benzodiazepine, z-drugs, and gabapentinoids; and, where applicable, concurrent prescription of benzodiazepines, z-drugs, and gabapentinoids.

DRP, drug-related poisoning; OAT, opioid agonist treatment.

prescribed benzodiazepines. This increased risk was specific to deaths from DRP, occurred during and after opioid substitution treatment, and was higher with doses of benzodiazepines above those recommended in the British National Formulary. Adjustment for measured possible confounding factors did not reduce these estimates of effect.

OAT reduces mortality risk; however, whether prescribed during OAT or in the 12 months after leaving OAT, benzodiazepines increased mortality risk from DRP by approximately 3-fold. We found no evidence of an interaction between the effect of benzodiazepine co-prescription on mortality risk and treatment period. This is probably because most opioid-dependent individuals continue using illicit opioids in the year following the end of a treatment episode [3]. We also found evidence that concurrent prescription of benzodiazepines was associated with longer OAT episodes. In general, longer OAT duration is associated with lower mortality. However, we found no evidence that the longer treatment duration associated with concurrent benzodiazepine prescription led to an overall reduction in mortality [3,7,19].

Z-drugs when co-prescribed with OAT were also associated with increased risk of mortality. This increased risk was seen both for ACM and DRP but showed no evidence of a dose–response effect. Co-prescription of gabapentinoids was associated with increased risk of ACM. This effect was substantially attenuated on adjustment for possible confounding factors.

## Comparison with other studies

Studies from North America have shown increased risk of both mortality from overdose and hospital attendance for overdose amongst patients prescribed both benzodiazepines and opioids [20,21]. These studies did not specifically examine effects in opioid-dependent individuals receiving OAT, and they acknowledged their limited ability to infer causality. A previous study from Scotland did examine the effects of benzodiazepine co-prescription amongst patients receiving methadone [10]. In keeping with our own findings, this study found no strong evidence of increased risk of ACM amongst co-prescribed patients, whereas an increase in DRP (HR 4.35 [95% CI 1.32–14.30], *p* < 0.05) was apparent in adjusted analyses. Only limited information on other patient characteristics was available in the Scottish study, constraining the ability to address confounding; possible dose–response effects were not assessed, and specific effects of concurrent prescription were not examined.

A recent small study from a single practice in London found that a high proportion of patients on OAT received a concurrent prescription of a benzodiazepine [13]. In keeping with our study, this study showed that concurrent prescription increased treatment duration. The overall effect on mortality of this increased treatment was not studied, though the authors suggested this would be positive. A recent Swedish register-based study examined effects on mortality amongst opioid-dependent individuals receiving OAT with co-prescription of benzodiazepines, z-drugs, and pregabalin [22]. This study included considerably fewer patients and had shorter follow-up than our own. Co-prescription of all these drugs was associated with increased risk of mortality, though for benzodiazepines this risk was stronger in relation to ACM than DRP. Dose–response effects were not examined, and adjustment for possible confounding was limited by a relative lack of information on other patient characteristics. This study did not consider effects on treatment duration. Two recent Canadian case–control studies considered the effects of concomitant gabapentinoid prescription amongst patients prescribed opioids for any reason other than palliative care or cancer and found evidence of increased risk of opioid-related death associated with both gabapentin and pregabalin prescription [23,24]. A recent research letter describes an increase in use of gabapentinoids in the US between 2002 and 2015 and also highlights increases in patients being prescribed gabapentinoids, opioids, and benzodiazepines concomitantly [25]. Pharmacovigilance around gabapentinoid prescribing has been increased in the US in response to concerns of harms associated with these medicines [26]. A further study also based on CPRD data also found evidence that prescribing of gabapentinoids increased in the UK between 1993 and 2017 [27].

The proportion of DRP deaths involving gabapentinoid use is increasing in the UK, and in most of these deaths opioid use is also recorded [28]. There is also animal evidence that gabapentinoids enhance the respiratory depressant effect of opioids [29].

## Strengths and limitations of this study

To our knowledge, our study is the largest and most detailed to date to examine the question of the effects of prescription of benzodiazepines, z-drugs, and gabapentinoids amongst opioid-dependent patients receiving OAT in primary care. Prospective observational studies based on administrative data provide the most robust means available to study this important clinical issue. Such studies are inevitably prone to bias, particularly confounding by indication. This is also true of our own study, though we adjusted for a wide range of possible confounding factors. Statistical methods based on propensity scores and the use of instrumental variables such as physician prescribing preference have been used in attempts to reduce this bias [30]. We have previously found that the former provide little benefit, as have others [15,31]. We were unable to identify an instrument suitable to allow the latter. We provide the strongest evidence currently available that benzodiazepine co-prescription in opioid-dependent individuals is of itself a cause of increased risk of death from DRP rather than a marker of other patient characteristics that influence risk of death. This evidence is less strong in relation to the increased mortality risk seen with co-prescription of z-drugs and gabapentinoids. Patients receiving OAT may obtain benzodiazepines, z-drugs, or gabapentinoids from non-prescribed sources, and we were unable to measure this. The resulting exposure misclassification would be expected to dilute any apparent mortality increase seen with co-prescription. Individuals in CPRD are broadly representative of the population of England [14]. As the phenomena we investigated are likely to be pharmacologically mediated, we think our findings may be applicable in other countries.

## Conclusions and policy implications

Our findings may have implications for clinical policy. Prescription of benzodiazepines to opioid-dependent patients during OAT or whilst still using illicit opioids should generally be avoided, other than when clinical judgement suggests that the benefits of this are likely to be greater than the harms. Clinical guidelines should emphasise this point. We found some evidence that guidelines may be having an impact, as the prevalence of co-prescription fell over our observation period. In contrast, in North America, though the prevalence of co-prescription is lower overall than in the UK, it appears to be increasing [32]. In the UK, some funding for primary care is linked to treatment quality indicators [33]. Benzodiazepine co-prescription with OAT could be used as a quality indicator in this way. Additional educational support to practitioners and psychological support to patients could also be offered in response to evidence of co-prescription. Periodic urine toxicology is widely used in OAT to monitor treatment compliance and could also be used to detect concomitant use of non-prescribed benzodiazepines, and additional support to reduce this offered. Our data provided some evidence that co-prescription of z-drugs in OAT patients may increase DRP risk such that continued recommendation against their co-prescription in guidelines, alongside the additional measures described above for benzodiazepines, seems appropriate. Our data did not show a significant association between co-prescription of gabapentinoids to patients receiving OAT and increased risk of DRP. These analyses were underpowered. Further analyses on the effects of gabapentinoid co-prescription based on large clinical databases are needed and will soon be feasible given increases in the coverage of clinical databases and apparent increases in the prescribing of gabapentinoids.

## Supporting information

**S1 Appendix. Summary of analyses and associated commands in Stata.**
(DOCX)

**S1 Patient Flowchart.**
(DOCX)

**S1 Protocol.**
(DOCX)

**S1 STROBE Checklist.**
(DOCX)

**S1 Table. Definition of drug-related deaths.**
(DOCX)

**S2 Table. Benzodiazepine and z-drug dose criteria.**
(DOCX)

**S3 Table. All-cause, drug-related, and non-drug-related mortality rates for covariates included in the analyses.**
(DOCX)

**S4 Table. Poisson regression analyses showing incidence rate ratios for ACM, DRP mortality, and non-DRP mortality according to co-prescribed medications.**
(DOCX)

**S5 Table. Adjusted all-cause, drug-related, and non-drug-related mortality incidence rate ratios for benzodiazepine, z-drug, and gabapentinoid exposure excluding first episode.**
(DOCX)

**S6 Table. Mortality rates and hazard ratios for DRP by co-prescription exposure to benzo-diazepines considering competing risk of death from other causes, z-drugs, or gabapenti-noids in people with opioid dependency in primary care.**
(DOCX)

## Acknowledgments

We are grateful for comments and contributions from patients and drug workers at Bristol Drugs Project (BDP) and Horizon, Blackpool, and in particular Rachel Ayres and Louisa Chowen at BDP for organising patient and public involvement.

## Disclaimer

The funder had no role in study design, data collection, the analysis and interpretation, or the writing of this report. The contents of this report do not necessarily reflect the views or stated position of National Institute for Health Research or the Department of Health.

## Author Contributions

**Conceptualization:** John Macleod, Matthew Hickman.

**Data curation:** Colin Steer.

**Formal analysis:** John Macleod, Colin Steer, Kate Tilling, Rosie Cornish, Matthew Hickman.

**Funding acquisition:** John Macleod, John Marsden, Tim Millar, John Strang, Matthew Hickman.

**Methodology:** John Macleod, Colin Steer, Kate Tilling, John Marsden, Tim Millar, John Strang, Matthew Hickman.

**Project administration:** Matthew Hickman.

**Writing – original draft:** John Macleod, Colin Steer, Kate Tilling, Matthew Hickman.

**Writing – review & editing:** Rosie Cornish, John Marsden, Tim Millar, John Strang.

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
