## [Decision Letter · Decision Letter 0]

6 Jul 2019

Dear Dr. MacLeod,

Thank you very much for submitting your manuscript "Prescription of benzodiazepines, z-drugs and gabapentoids and mortality risk in people receiving opioid substitution therapy: observational study based on the UK Clinical Practice Research Datalink and Office of National Statistics death records" (PMEDICINE-D-19-02004) for consideration at PLOS Medicine. 

Your paper was discussed among the editorial team, evaluated by the guest editors for the special issue, and sent to independent reviewers, including a statistical reviewer. The reviews are appended at the bottom of this email and any accompanying reviewer attachments can be seen via the link below:

[LINK]

In light of these reviews, we will not be able to accept the manuscript for publication in the journal in its current form, but we would like to invite you to submit a revised version that fully addresses the reviewers' and editors' comments. You will appreciate that we cannot make a decision about publication until we have seen the revised manuscript and your response, and we expect to seek re-review by one or more of the reviewers. 

We hope to receive your revised manuscript within two weeks. Please email us (plosmedicine@plos.org) if you have any questions or concerns.

Please let me know if you have any questions. Otherwise, we will look forward to receiving your revised manuscript shortly. 

Sincerely,

Richard Turner PhD, for Philippa Berman, MBBS

rturner@plos.org

In the metadata, please briefly mention the restrictions on access to CPRD data that interested readers would encounter. 

Please convert your abstract to PLOS Medicine style. The final sentence of the combined "methods and findings" subsection should quote the study's main limitations. 

Please include brief demographic and follow-up details in your abstract; and add p values, where available, alongside CI. 

Please begin the "conclusions" subsection of your abstract with "In this study ..." or similar, and adopt the past tense when describing findings.

After your abstract, we will need to ask you to add a new and accessible "author summary" section in non-identical prose. You may find it helpful to consult one or two recent research papers published in PLOS Medicine to get a sense of the preferred style. 

Early in your methods section, please state whether or not the study had a protocol or prespecified analysis plan, and if so add the relevant document as a supplementary file. Please highlight all analyses that were not prespecified.

Please briefly state the position regarding ethics approval in your methods section. 

The first paragraph of the discussion section of your main text should summarize the study's findings, and we suggest beginning this with "We found that ..." or similar. If the subheading "Principle findings" remains, please adapt it to "Principal findings". 

The discussion section should also include a discrete paragraph summarizing the study's limitations. Mitigating factors can be discussed in a subsequent paragraph. 

Where you make claims such as "Our study is the largest ...", as you do in the discussion section, please add "to our knowledge" or similar. 

Please include p values, where appropriate, alongside CI throughout your main text.

Please substitute "sex" for "gender" throughout your paper, where appropriate. 

Throughout your article, please call out references with numbers in square brackets preceding punctuation, e.g. "... are in treatment [1].".

Throughout the paper, please substitute "p<0.001" or an exact value for "p<0.0001".

Please ensure that your reference list conforms with journal style. Up to six author names should be listed, followed by "et al." where appropriate. Journal names should be abbreviated consistently, and all italics should be removed. 

Please add a completed checklist for the most appropriate reporting guideline, which we suspect will be STROBE or RECORD, as a supplementary file, and refer to this in the methods section. Individual items should be referred to by section and paragraph number rather than by line or page numbers, as the latter generally change upon publication. 

Comments from academic editor:

1. The authors' exclusions result in a cohort that is 25% of the original sample. This has implications for generalizability and should be discussed thoughtfully. Second, I am surprised that 50% of the excluded patients were excluded because they were being prescribed buprenorphine or methadone "for pain relief". This proportion seems large. Compared to other large pharmacoepidemiological databases in the US or Europe, is this what one would expect? Or is there just a lot of misclassification going on? This should also be discussed thoughtfully. Sensitivity analyses would also go a long way toward reassuring readers that the estimated associations are robust.

2. Why did treatment episodes include 1 year after the prescription ended? This seems arbitrary and should perhaps be probed with some sensitivity analyses. (The authors select a different cutoff of 14 days for the BZD/z-drug exposure, reasoning that those medications are associated with a "shorter recommended treatment duration". But recommended treatment duration with methadone/buprenorphine is presumably lifelong. So why 28 days?) In addition, I am a little confused about how this worked with patients who had consecutive episodes. If a new episode was defined as a subsequent prescription >28 days after a previous prescription, what if someone had a prescription 45 days after a previous prescription ended? They would still be within the 1-year window, so they would be classified as "in treatment" and this time in treatment would be associated with the first prescription? 

3. The authors do not appear to have made any effort to ensure that treatment episodes (of OST or the co-prescribed BZD/z-drug/gabapentinoids) are "new" treatment episodes. 

4. Please justify in the text the use of linear regression (rather than, say, Poisson, negative binomial, two-part models, etc) in the analysis of treatment duration. The authors do not appear to have used robust standard errors so the estimates are unlikely to be robust to a wide range of distributional assumptions. 

5. After reading through the methods several times, it is still a little unclear to me exactly what models were used here:

(a) "Where a main effect of co-prescription was suggested by these analyses we investigated whether this effect was modified by an interaction with OST treatment period (i.e. on or off treatment)." 

(b) "For benzodiazepine or z-drug dose effects (2 degrees of freedom), a linear trend was also fitted to examine the effect of dose, with “high” doses ascribed based on a daily dose above the maximum recommended in the BNF (see Web table 1)."

(c) "We then assessed the overall effect of concurrent prescription (allowing for any effect on treatment duration) on mortality risk, using Cox proportional hazards survival analysis."

It would be helpful if the regression equations corresponding to these sentences were included (eg., in an Appendix).

6. The specificity of the findings is unclear. This would be strengthened by the use of an ancillary analysis in which comparable analyses were done with other sedating medications that are commonly co-prescribed in this setting (eg. antidepressants, antipsychotics) as well with other non-sedating medications that are commonly co-prescribed but would not be expected to increase DRP deaths (eg., NSAIDs). This type of analysis is commonly referred to as a "falsification test" (eg., Prasad & Jena, JAMA 2013;309:241-2) or the use of "negative controls" (eg., Arnold et al. Epidemiology 2016;27:637–641)).

MINOR COMMENTS:

7. I would recommend using the term "opioid agonist treatment", or some other non-stigmatizing term, rather than "opioid substitution therapy". See Wakeman SE. J Addict Med 2017;11: 1–2.

8. The authors indicate they studied 10 benzodiazepines, 3 z-drugs, and 2 gabapentinoids. These should be named.

9. I understand that the CRPD initial cohort received a prescription between 1st January 1998 and 31st July 2014. In the Results section, the authors should provide the respective dates for the final analytic sample.

10. More detail should be provided in the summary statistics paragraph: "These data contained 29,540 treatment episodes: 17,391 methadone (median treatment duration, XXXX; range, XX-YY; interquartile range, XX-YY), 9180 buprenorphine (median treatment duration, XXXX; range, XX-YY; interquartile range, XX-YY) and 2969 episodes relating to more than one drug. Total follow-up time was 36,126 person years during which there were 17,139 episodes of benzodiazepine treatment, XXXX episodes of z-drug treatment, and YYYY episodes of gabapentinoid treatment. Benzodiazepines were prescribed during 10,022 OST episodes in 7059 patients. Z-drugs were prescribed during 4279 OST episodes in 2822 patients. Gabapentoids were prescribed during 1827 OST episodes in 1281 patients."

11. The authors indicate "mean follow-up per patient of 3.4 years" but this should be expressed as "median (IQR; range)". 

12. Throughout the manuscript the authors refer to "gabapentoids". These should be replaced with "gabapentinoid". 

13. The literature review is incomplete. Several studies of co-prescribing in this area have been published, but the authors neither cite nor interact meaningfully with this literature (eg., discussing the strengths of their study relative to the others).

Gomes T et al. PLoS Med. 2017 Oct 3;14(10):e1002396.

Gomes T et al. Ann Intern Med 2018;169:732-4

Johansen. JAMA Intern Med. 2018 Feb; 178(2): 292–294.

Peckham et al. Drug Saf 2018;41(2):213-228

Montastruc et al. JAMA. 2018;320(20):2149-2151.

Comments from the reviewers:

*** Reviewer #1: 

The authors present a cohort study using administrative data from 1998 - 2014 from the United Kingdom examining over 12,000 patients receiving opioid substitution treatment (OST) and their duration of OST use and mortality in relation to use of benzodiazepines, z-drugs (zaleplon, zolpidem, and zopiclone), and gabapentoids. The findings of the study most strongly suggest an increased duration of OST associated with concurrent benzodiazepine use but also a significantly increased risk of mortality relative to non-use. A few comments and questions are noted below: 

1. It's unclear whether the cohort was a 'new user' cohort (i.e. relatively new use of OST). If it was a new user cohort, please include the definition of 'new user'. If not, would it be possible to provide a sense of how long they had being OST prior to the cohort entry / start date? As the authors may be aware, prevalent user designs may be subject to various sources of bias (https://www.ncbi.nlm.nih.gov/pubmed/14585769). On a related note, how was the cohort entry date / start date defined? Was it the first instance of OST use observed during the study period? Please explicitly define. 

2. For the exposures of interest, these drugs are often used on an as needed (prn) basis and 14 days was assumed to be the minimum gap between prescriptions for purposes of estimating use (page 8). Was this an arbitrary decision? How was occasional use handled (e.g. some people may spread the use of 30 pills of benzodiazepines over 90 days)?

3. Was there a single primary endpoint of interest? There were numerous exposures, definitions of exposure duration, and outcomes. As a result, quite a few statistical tests were conducted. How was the issue of multiple hypothesis testing handled?

4. Were comorbidities assessed prior to cohort entry (i.e. the start date) or were 'new' comorbidities identified during follow-up included (page 8)? The use of time varying covariates in this regard is a bit confusing. Please explain. 

5. The definition of death involves multiple data sources (page 9). Has the outlined approach to defining death been validated? If so, how accurate is death using the approaches taken? Also, have there been any validation studies of the outcome of drug-related death poisoning? 

6. Linear regression was used to examine the association between concurrent use of an exposure of interest and OST. Was a time-to-event analysis such as a survival model considered for this assessment? Were competing risks considered? 

7. It would be helpful to more fully describe how selection bias was handled in this study. For example, patients not using a benzodiazepine may be considerably different than those who use a benzodiazepine. Such information has not been provided. Use of time-varying covariates in a survival model does not obviate selection bias. Further, timing of as needed drugs such as benzodiazepines may reflect critical events in a patient's life that lead to the use of such medications. The critical events may have a stronger effect on the risk of mortality than the actual exposure being assessed (i.e. the exposure to a drug such as a benzodiazepine at a particular time may mask a critical event that may not be captured by the available data). It would be helpful for the authors to highlight how such temporal confounding was handled, if at all. 

*** Reviewer #2: 

This is a well-conducted epidemiological study on the associations between prescription of benzodiazepines, z-drugs and gabapentoids and mortality risk in people receiving opioid substitution therapy based on UK's CPRD primary care database and death records from the Office of National Statistics. The study design, datasets, statistical methods and analyses, presentations (tables and figures) and interpretations of results are mostly adequate and of a good standard. However, there are still a few important statistical issues needing attention.

1) The key conclusion of the study is that Co-prescription of benzodiazepine is associated with a three-fold increased risk of DRP (drug-related poisoning mortality) as shown in table 3. This was done by the Cox model. However, in this case all cause mortality becomes a competing risk that could potentially hinder or modify the chance of DRP, therefore it would be appropriate to estimate the risk of DRP within the competing risk framework. Unfortunately the authors didn't address this issue in the paper.

2) Table 2 on treatment durations. In the second column, median duration was presented which suggested that the distributions of the duration were not normally distributed. Is this right? If so, then these durations as outcome should be transformed (e.g., log-transformation) into normal variables before they can be used in linear regressions. Can authors check, report and correct as appropriate?

3) Table 1. Follow-ups (years) should be summarised as median and IQR rather than mean and SD as they are skewed data.

*** Reviewer #3: 

This paper is important as it addresses a question that has, and should be, high on the agenda for policy makers and clinical practitioners managing people who have used drugs.

It has been difficult in the past to unravel the interactions of drugs and their, often fatal, effects. Using administrative data is an innovative way to provide some clarity and although the authors acknowledge the incomplete nature of their enquiry into this complex area they have opened up some very important leads and articulated the research questions that need to be answered.

Are benzodiazepines prescribed because methadone doses are too low? This is a common problem in clinical practice and diazepam particularly, but also other sedatives such as Z drugs, are prescribed or obtained illegally to supplement the effects of inadequate opiate substitute treatment.

People on <20mg methadone excluded. This clearly is a dose way below the recognized therapeutic level but isn't 30 or 40mg in the same range?

Are these patients using non prescribed BDZs. How is it known that the BDZ present at death isn't non prescribed?

Mortality showed regional variation, is there any explanation for this variation and is it associated with coprescribing?

With evidence of a dose response effect can it ne made clearer how strong this was? Is it not true that there would be and expectation that it would be significant?

Why not include other drugs with an equal potential for additional respiratory depression such as SSRIs and those with QRS complex impairment? The former are widely used to supplement opiate substitute treatment and to complement the management of the symptomatology around a lifestyle of opiate use. 

What is the mechanism for association of co prescription of benzodiazepines and methadone and DRP? Is it possible to comment or even speculate on whether or not this is cumulative respiratory depression or if, as had been suggested, that co prescribed drugs has a behavioural effect in inducing confusion/recklessness or misjudgement of doses?

We already know that deaths are associated with multiple drugs at toxicology but the causation is complex and less well understood.

Research is needed into the reasons why overdose can occur when it hasn't before and the role of other factors such as COPD and GI (liver primarily) disorders, reference for example;

Lu Gao J. Roy Robertson Sheila M. Bird Non drug-related and opioid-specific causes of 3262 deaths in Scotland's methadone-prescription clients, 2009-2015 Drug and Alcohol Dependence Volume 197, 1 April 2019, Pages 262-270

Is this subset a higher risk group with different characteristics? It is recognised that drug users are not a homogeneous group with varying behaviour patterns suggesting an increased risk

Mortality risk during and after opioid substitution treatment: systematic review and meta-analysis of cohort studies

BMJ 2017; 357 doi: https://doi.org/10.1136/bmj.j1550 (Published 26 April 2017)

Cite this as: BMJ 2017;357:j1550

The conclusion that benzodiazepines should not be coprescribed is a sweeping statement based on results that don't necessarily take into account the best intentions of clinical services that are trying to balance the needs for symptomatic treatment of a distressed caseload.

Avoiding one drug (such as diazepam in this case) often leads prescribers to look for an alternative, which may be equally toxic.

The other, perhaps more recent, problem is the escalation in availability of synthetic or illegally sourced benzodiazepines. Supplies of etizolam and alprazolam have distorted the marked and caused enormous self medication problems leading to increased risk of death. One response to this has been to accommodate these needs by prescribing benzodiazepines.

Perhaps the underlying problem is inadequate dosing of methadone or buprenorphine and it might be useful to say this.

***

[LINK]

---

## [Decision Letter · Decision Letter 1]

30 Sep 2019

Dear Dr. MacLeod,

Thank you very much for re-submitting your manuscript "Prescription of benzodiazepines, z-drugs and gabapentoids and mortality risk in people receiving opioid substitution therapy: observational study based on the UK Clinical Practice Research Datalink and Office of National Statistics death records" (PMEDICINE-D-19-02004R1) for review by PLOS Medicine.

I have discussed the paper with my colleagues and the academic editor and it was also seen again by reviewers. I am pleased to say that provided the remaining editorial and production issues are dealt with we are planning to accept the paper for publication in the journal.

[LINK]

We look forward to receiving the revised manuscript by Oct 07 2019 11:59PM. 

Sincerely,

Richard Turner,PhD

Senior Editor 

PLOS Medicine

plosmedicine.org

Requests from Editors:

Abstract – where there are 95% values for quantifiable data, please also add p values (also in main text and tables, where necessary). Even though the abstract does not exceed the requested 500 words, that doesn’t need to be a target and I find it quite long and wonder if it could be trimmed for conciseness? 

Abstract – as this isn’t a trial, please be cautious to avoid causal language…for example in the abstract “In this study, co-prescription of benzodiazepine was associated with increased risk of DRP in opioid dependent individuals both on and off OAT. Co-prescription of z-drugs and gabapentinoids was also associated with increased mortality risk, however evidence for causality was less strong.” Please avoid the word evidence and also vague language…an association is either significant or not and if not, remove – or say no association. Also line 312 (causal effect). 

Author summary – thanks you for adding this; it needs to be in a bullet point format. 

Line 259 – “was not strongly associated with age” – please be specific in terms of significance of association – it is, or isn’t. 

Line 423 (should) please say ‘may be ‘ and would bit be more accurate to say applicable in other countries instead of externally valid? 

Line 428 (Our findings have clear implications for clinical policy) – please say ‘may have’ and remove ‘clear’

Line 442 (Our data did not provide strong evidence that co-prescription of gabapentinoids to patients receiving OAT causes increased risk of DRP) – please be clear – it did, or didn’t…

STROBE – thanks for providing. Instead of METHODS THIRD TO FIFTH PARAGRAPH (for example), please say Methods Paragraphs 3-5 and adjust throughout. Also please remove the word ‘Done’.

Comments from Reviewers:

Reviewer #1: While the authors have dutifully responded to the many comments of the editors and reviewers, I am not convinced that the validity of the study findings are high enough to warrant publication in PLOS Medicine. Several outstanding issues have either not been addressed adequately or simply cannot be addressed well given the nature of the study design: 

1. Generalizability: The academic editor raised the issue of a very high exclusion rate (i.e. a study cohort that is 25% of the original sample). While the authors provide some explanations for the exclusions, the level of misclassification is unclear and a validated approach to arriving at the study sample would have been extremely helpful. Simply accepting a possibly high level of misclassification doesn't seem adequate. Some more 'homework' needs to be done in this regard. 

2. Establishing a 'new user' cohort: New user designs are known to minimize bias, however the authors acknowledge that this study design is virtually impossible with the data sources available. It may be that 'chronic' OAT users may be more likely to use one of the study drugs than another. The use of time-dependent exposure assessment may help to mitigate some of the potential biases here, but it still may be problematic. 

3. Temporal confounding and specificity of findings: Perhaps most concerning is the possibility of temporal confounding as it relates to protopathic bias. The issue here is that the initiation of a drug (e.g. a benzodiazepine) may 'unmask' an underlying problem (either related or unrelated to the drug being initiated) that then leads to an outcome such as death or overdose. The use of 'falsification tests' or 'negative controls', as suggested by the academic editor may help understand this issue better. The use of antidepressants / antipsychotics and NSAIDs was suggested by the academic reviewer. The authors did not examine the effects of antidepressants / antipsychotics since they felt there may be considerable confounding by indication. I suppose, perhaps to a lesser extent, the same argument could be made for benzodiazepines. The authors did, however conduct such analyses using antimicrobials and NSAIDs. The findings highlight the issue of bias as statistically significant associations between antimicrobials and several of the outcomes of interest were observed. The risk estimates for the association between NSAIDs and the outcomes of interest were also elevated but not statistically significant. These findings raise serious concerns about the validity of the observed association between benzodiazepines and the selected outcomes. The authors seem to dismiss the associations between antimicrobials and the selected outcomes by suggesting these are driven by confounding by indication and fail to consider that this may also be what's driving the observed associations between benzodiazepines and the selected outcomes. 

For the above key reasons, I question the validity of the findings of this study. 

Reviewer #2: Thanks authors for their effort to improve the manuscript. The response and revision are mostly satisfactory. However, still one minor issue remaining. In the revised table 1, could authors please remove the row with Mean follow-up years (SD)? For skewed data, it doesn't make sense and is inadequate to use mean and SD. Median and IQR are sufficient and adequate in this case.

Reviewer #3: I am happy that the authors have considered and responded to my suggestions and concerns. I understand that many of my comments were outwit the range of the study and my expectations were, for that reason, unrealistic. I do think that more research is required to illuminate some of the unanswered questions but that is clearly beyond the scope of this study.

I am still worried about the statement in the abstract and text that prescribers should avoid prescribing benzodiazepines completely. This is an unrealistic expectation in an era of enhanced Harm Reduction when availability of illegal benzodiazepines and self medication is at an all time high. I accept that BDZs and other coprescribed drugs may increase the risk of death but there must be a balance of risks present (as there is with many drugs that are commonly prescribed in patients with serious disease). For an influential group to make a uncompromising statement is likely to cause unintended consequences. Guidelines will cite it and conclude that prescribing any BDZ is outwit normal practice and any strategy to reduce harm by prescribing a (relatively safer) benzodiazepine will be inhibited.

Could the authors draw attention to the risk they have identified and suggest less binary solution?

[LINK]

---

## [Editor Report · Decision Letter 2]

24 Oct 2019

Dear Dr. MacLeod, 

On behalf of my colleagues and the academic editor, Dr. Alexander Tsai, I am delighted to inform you that your manuscript entitled "Prescription of benzodiazepines, z-drugs and gabapentoids and mortality risk in people receiving opioid substitution therapy: observational study based on the UK Clinical Practice Research Datalink and Office of National Statistics death records" (PMEDICINE-D-19-02004R2) has been accepted for publication in PLOS Medicine. 

PRODUCTION PROCESS

PRESS

PROFILE INFORMATION

Thank you again for submitting the manuscript to PLOS Medicine. We look forward to publishing it. 

Best wishes, 

Richard Turner, PhD

Senior Editor 

PLOS Medicine

plosmedicine.org